

# Drift simulation of MH370 debris using supersensemble techniques

Eric Jansen[1], Giovanni Coppini[1], and Nadia Pinardi[2,1]

[1]Euro-Mediterranean Center on Climate Change (CMCC), Lecce, Italy
[2]Department of Physics and Astronomy, Alma Mater Studiorum University of Bologna, Bologna, Italy

*Correspondence to:* Eric Jansen (eric.jansen@cmcc.it)

**Abstract.** On the 7th of March 2014 (UTC), Malaysia Airlines flight 370 vanished without a trace. The aircraft is believed to have crashed in the southern Indian Ocean, but despite extensive search operations the location of the wreckage is still unknown. The only part of the aircraft that has been recovered so far is a small piece of the right wing. It was discovered 17 months after the disappearance on the island of Réunion, approximately 4,000 km from the assumed crash site.

This paper presents a numerical simulation using high resolution oceanographic and meteorological data to predict the movement of floating debris from the accident. It combines multiple model realisations into a superensemble, and includes the discovery of debris on Réunion to improve the final result. The superensemble is used to predict the distribution of debris at various moments in time.

The results for the initial probability density show good agreement with the current underwater search area. Results at later times show that the most probable locations to discover washed up debris are along the African west-coast and the southeast of Australia. The debris remaining at sea from late 2015 is spread out over a wide area and its distribution changes only slowly.

## 1   Introduction

Modern sea situational awareness technologies make use of both real time information and advanced, long term reconstructions of the ocean state. Among many other applications, the ocean reconstructions allow the study of transport and dispersal scenarios for objects and pollutants at sea. Such studies are key to preparing better emergency response management plans and performing post-crisis assessments. This paper shows the use of long term global ocean and weather reconstructions to produce probability distributions for the surface drift of aircraft debris originating from an accident in the southern Indian Ocean.

Malaysia Airlines flight 370 (MH370) was a scheduled passenger flight from Kuala Lumpur, Malaysia, to Beijing, China. The Boeing 777-200 ER aircraft carrying 239 passengers and crew disappeared less than an hour after take-off. Air traffic control (ATC) lost voice and radar contact with the flight at 17:22 UTC on the 7th of March 2014, while the aircraft was over the Gulf of Thailand. Initially it was assumed that the flight had crashed, but analysis of military radar data showed that the plane had deviated from its planned flight path and returned towards Malaysia. The aircraft continued flying in a westward direction and eventually exited the radar coverage at 18:22 UTC (Safety Investigation Team for MH370, 2015).

While most on-board communication equipment was inoperable, minimal communication between the aircraft's satellite terminal and the satellite network continued until 0:19 UTC. Analysis of this communication (Ashton et al., 2015) concluded that the aircraft changed course towards the south and continued in this direction until fuel ran out. Based on this information,



various search areas have been defined in the southern Indian Ocean in an effort to locate the wreckage (ATSB, 2015). These areas straddle the arc of constant distance to the satellite at the time of the final communication. The most important areas are shown in Fig. 1. The *wide search area* is the envelope of possible locations, but due to its size a full search of this area is not feasible. Motivated by the most probable flight path, underwater searches for the wreckage have concentrated in the southern

part of this area. However, the wreckage has not been located. The only part of the aircraft that has been recovered so far is a piece of the right wing, approximately 2 m in size, that was found on the beach of the French island of Réunion on the 29th of July 2015.

The following sections will presents a numerical simulation that uses all available information about the crash site and the discovery on Réunion to predict the distribution of floating debris from the accident. The paper is organised as follows: in

Sect. 2 the drift modelling and the treatment of unknown initial conditions is discussed; Sect. 3 shows the predictions of the model in terms of a time-series of debris probability maps; and finally, Sect. 4 presents the conclusions of this study.

## 2  Modelling

### 2.1  Drift trajectories

Drift trajectories are modelled using an ensemble of particles drifting on the ocean surface. Due to differences in their initial

positions and due to random motion these particles will slowly diverge over time. Each trajectory represents a possible path of the object being studied.

In this model the displacement $\mathrm{d}\boldsymbol{x}_i$ of particle $i$ in an infinitesimal time interval $\mathrm{d}t$ is given by:

$$\mathrm{d}\boldsymbol{x}_i = [\boldsymbol{c}(\boldsymbol{x}_i,t) + L\boldsymbol{w}(\boldsymbol{x}_i,t)]\,\mathrm{d}t + \boldsymbol{n}. \tag{1}$$

Here $\boldsymbol{c}(\boldsymbol{x}_i,t)$ and $\boldsymbol{w}(\boldsymbol{x}_i,t)$ are the ocean current and wind forcing fields, interpolated at position $\boldsymbol{x}_i$ and time $t$. The constant $L$

is the wind drag coefficient. The term $\boldsymbol{n}$ parameterises the incoherent turbulent motion of the particle. Its components $n_j$ are random variables, distributed according to:

$$n_j \sim \mathcal{N}\left(0, \sqrt{2K_h\Delta t}\right) \qquad\qquad K_h = 2\,\mathrm{m^2s^{-1}} \tag{2}$$

with $\mathcal{N}(\mu,\sigma)$ the normal distribution and $\Delta t$ the length of an integration timestep. The constant $K_h$ is the turbulent diffusion coefficient. The value of this coefficient is taken from De Dominicis et al. (2013).

The particles in the model are advected by integrating Eq. (1) numerically using the 4th order Runge-Kutta method.

### 2.2  Superensemble

To account for the fact that the location of the wreckage and the wind drag coefficient are not accurately known, the results presented here use multiple realisations of the model, with varying initial conditions and parameters. The different realisations are then combined into a superensemble, inspired by the use of multimodel forecasts in meteorology (Krishnamurti et al.,

1999).



The superensemble probability $P(\boldsymbol{x}, t)$ of finding aircraft debris in a location $\boldsymbol{x}$ at time $t$ is expressed as a sum over all model realisations $k$ and their individual probabilities $P_k$ as:

$$P(\boldsymbol{x}, t) = \sum_k \alpha_k P_k(\boldsymbol{x}, t), \tag{3}$$

with $\alpha_k$ the model weight coefficient. The probability predicted by the $k$th model realisation can be defined as:

$$P_k(\boldsymbol{x}, t) = \frac{1}{N_k} \sum_i \delta(\boldsymbol{x} - \boldsymbol{x}_{k,i}(t)), \tag{4}$$

with $N_k$ the total number of particles and $\delta$ a function that equals 1 below a certain distance and 0 otherwise.

### 2.2.1 Model coefficients

In order to obtain meaningful results from the superensemble, the coefficients $\alpha_k$ are chosen to represent the likelihood of a particular model realisation $M_k$, given the fact D that debris washed up on Réunion. Using Bayesian inference $\alpha_k$ can be expressed as:

$$\alpha_k = \mathcal{P}(M_k|D) \propto \mathcal{P}(D|M_k) \times \mathcal{P}(M_k). \tag{5}$$

Here $\mathcal{P}(D|M_k)$ can be obtained from Eq. (4). Particles are selected if they appear within $100\,\mathrm{km}$ of Réunion in the 3 months leading up to the debris discovery. The prior model likelihood $\mathcal{P}(M_k)$ is chosen inversely proportional to the model variance:

$$\mathcal{P}(M_k) = \frac{1/\sigma_{x,k}^2}{\sum_n 1/\sigma_{x,n}^2}. \tag{6}$$

The variance is evaluated at time $t = 90$ days to capture the variability of the different model realisations without particles stranding on the coastline.

### 2.2.2 Initial conditions

The superensemble is initialised with 30 model realisations, each containing 5,000 particles. The realisations contain all possible combinations of 6 different wind drag coefficients and 5 starting locations. The wind drag coefficients range from 0 to 2.5 % in increments of 0.5 %, spanning roughly the range from no wind drag at all to a drag similar to that of a small boat (Allen and Plourde, 1999). The starting locations are obtained by dividing the wide search area (see Fig. 1) in 5 sections of equal size, along the arc of the final satellite communication.

### 2.3 Forcing field data

The oceanographic and meteorological data for the simulation are provided by the European Copernicus Marine Environment Monitoring Service. The ocean currents at the surface level are extracted from the global ocean analysis and forecast[1] data,

---

[1] Dataset GLOBAL-ANALYSIS-FORECAST-PHYS-001-002





which provide a 3-dimensional ocean reconstruction with global coverage at a spatial resolution of $1/12°$. The temporal resolution for the period of the simulation is 2-hourly, except for the period of 8–18 March 2014 where the daily average is used. The wind forcing uses the global near real time ocean wind observations[2], which consist of satellite scatterometer observations combined with the operational wind analysis from the European Centre for Medium-Range Weather Forecasts

(ECMWF). The wind fields are measured as a 6-hour time average with a spatial resolution of $1/4°$. The wind speed is defined at a reference height of 10 m.

## 3   Results

Figures 2 and 3 show the distribution for floating MH370 debris in the southern Indian Ocean at various times. The amount of debris predicted in a certain location is expressed as the number of simulated drifting particles in an area of $1° \times 1°$. The

number of particles is normalised to that used in a single model realisation, i.e. $\sum_k \alpha_k = 1$. Areas containing less than 2 drifters per square degree are not shown.

    While the weights are calculated based on the predicted distribution at the time of the discovery on Réunion, the superensemble can be used to predict the debris distribution at any point in time. One distribution that is of particular interest is the initial distribution, shown in the top left panel of Fig. 2. It shows that the origin of the debris on Réunion, and therefore the location

of the crash site, is more likely to be in southern half of the wide search area.

    The time evolution of the debris distribution shows that debris originating from the southern part of the wide search area initially drifts towards the east and may have reached the coast of Australia from September 2014 onwards. However, the majority of the debris eventually drifts in a westward direction towards Madagascar, possibly reaching land in October 2014. As early as January 2015 debris is predicted close to the island of Réunion, where a part of the wing was eventually found in

August 2015. The debris that remains at sea from late 2015 is spread out over a large area and its distribution changes only slowly.

    While debris may wash up on shore in various locations, it is important to note that there are many other factors that determine whether the debris will be found, recognised and reported to the proper authorities. In sparsely populated areas it may take a long time before debris is discovered, while in developing countries a discovered piece of debris may be discarded

because people are not aware of the missing aircraft. It is therefore not unlikely that debris has washed up in other locations, but remained unreported.

## 4   Conclusions

The availability of high resolution oceanographic and meteorological data is key to adequately responding to accidents and emergencies at sea. The case of MH370 especially underlines the importance of data that provide global coverage. With the

operational range of modern airliners in excess of 10,000 km, accidents may happen in remote areas that are deprived of

---

[2]Dataset WIND-GLO-WIND-L4-NRT-OBSERVATIONS-012-004





most means of communication. In such cases the meteo-oceanographic data might be the only source of information on the circumstances of the accident and the conditions afterwards.

The results presented in this paper show that the discovery of debris on Réunion is in good agreement with the satellite data analysis and the current underwater search area that was derived from it. Due to the large uncertainties associated with the modelling of drift trajectories, the results unfortunately do not permit the exclusion or confirmation of any part of the search area.

Based on the particle distribution, the most probable locations to discover further washed up debris from flight MH370 are the northwestern part of Madagascar; the islands Réunion and Mauritius; the Tanzanian coast; and the Australian coast south of Perth. The time evolution shows that from late 2015 onwards the floating debris that remains at sea is spread out and its distribution changes only slowly. The probability of new debris washing up in the future is therefore quite low.

The results obtained for MH370 show that the superensemble method described in this paper is well-suited for drift simulations where additional observations are available. By adjusting the weights of the superensemble members it is straightforward to incorporate information such as the debris discovered on Réunion into the simulation. In a conventional ensemble this would be very difficult. Moreover, the proposed method can be easily extended to include new information. As long as the superensemble is of sufficient size, only the model coefficients $\alpha_k$ have to be recalculated.

*Acknowledgement.* This study has been conducted using the Copernicus Marine Service Products (http://marine.copernicus.eu).



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

## Appendix A: Mozambique debris

On the 27th of February 2016, a possible second piece of debris was discovered in the channel between Madagascar and the African mainland. The piece, with a length of around 1 m, was found stranded on a sandbank off the coast of Vilankulo, Mozambique ($22°0'$ S, $35°19'$ E; see Fig. 4).

At the time of publication of this paper it has not yet been confirmed whether this debris is related to flight MH370 or not. However, it is worth noting that the location of the discovery is in good agreement with the results presented here. In Fig. 3 it can be seen that the simulation predicts possible debris arriving on the coast of Mozambique during the second half of 2015.

Assuming that the newly found debris is indeed from flight MH370, it may be included in the superensemble simulation by updating the model weight coefficients $\alpha_k$. In this case $\mathcal{P}(\mathrm{D}|\mathrm{M}_k)$ in Eq. (5) becomes a product of multiple discoveries $\mathrm{D}_l$:

$$\mathcal{P}(\mathrm{D}|\mathrm{M}_k) = \prod_l \mathcal{P}(\mathrm{D}_l|\mathrm{M}_k). \tag{A1}$$

Calculating the probability of debris in Mozambique predicted by each model realisation proceeds analogously to the calculation for Réunion. In this case particles are selected if they appear within 50 km from the town of Vilankulo at any point in time.

Figures 4 and 5 show the results of the updated simulation. Including the new information leads to a slightly faster westward drift of the particles. In addition, the most probable location of the crash site is shifted towards the north. Including the possible new discovery in Mozambique does not significantly alter the conclusions of this paper.



**Figure 1.** Estimated flight path of flight MH370 based on military radar and satellite data analysis. Also indicated are the search areas and the island of Réunion.





**Figure 2.** Floating aircraft debris probability density between March and November 2014.







**Figure 3.** Floating aircraft debris probability density between January 2015 and January 2016.







**Figure 4.** Updated version of Fig. 2, taking into account both the debris found on Réunion and the possible new discovery in Mozambique. The location of the discovery in Mozambique is indicated in red.





**Figure 5.** Updated version of Fig. 3, taking into account both the debris found on Réunion and the possible new discovery in Mozambique. The location of the discovery in Mozambique is indicated in red.