# Peer review of "Drift simulation of MH370 debris using supersensemble techniques"

_Natural Hazards and Earth System Sciences, 2016_

## Referee Comment (RC1) · Anonymous Referee #1 · 14 Apr 2016

REVIEW OF Drift simulation of MH370 debris using superensemble techniques Author(s): E. Jansen, G. Coppini and N. Pinardi

This manuscript is a very important contribution to Natural-Hazards-And-Earth-System-Sciences and is entirely suitable for publication. The paper adds new knowledge to the overall body of scientific understanding regarding particle tracking in the environment. The authors, especially the second and third ones, are very well recognized and highly regarded researchers in the subject area of this manuscript. The authors build on their previous work with lagrangian modeling and employ well established meteorological (ecmwf) and oceanographic (European Copernicus Marine Environment Monitoring Service) data sets in their analysis. The use of drift simulations will take on especial importance as air and sea travel continues to rise and world events become more dangerous putting more of our population at risk. The research results

presented are highly original. The approach taken in the conduct of this research will set a standard for future drift modeling.

I believe the paper is free of errors in logic; their case is very well presented and made.

Some comments: 1. I think that the appendix should be moved into the main body of the manuscript. It is possible that the Mozambique debris may not be related to MH370. But the method of including it in the analysis is important for the readers. In fact there are other announcements of possible debris. It would be so good to include them all.

http://www.ibtimes.com/flight-mh370-update-chinese-vessel-resume-search-recover-lost-towfish-amid-suspected-2350977  http://www.livescience.com/54158-debris-found-from-mh370-malaysia-plane.html  http://www.inquisitr.com/2941135/malaysia-airlines-flight-mh370-new-debris/  http://www.inquisitr.com/2968872/malaysia-airlines-flight-mh370-inside-plane-debris/  http://www.inquisitr.com/2957754/malaysia-airlines-flight-mh370-planes-interior-debris/

2. It would be interesting if the authors could discuss a backtracking approach. Release particles from Reunion Island and see where they came from. Play the currents back in time.

3. Why is the approach called a "superensemble" as opposed to an ensemble? What makes is super?

4. It seems to me that there has been much published about the debris field. Has all the literature been reviewed for this manuscript?

5. I like the figures. I think Figure 1 should contain labels for Mozambique and Madagascar. Others?

---

## Referee Comment (RC2) · Anonymous Referee #2 · 20 Apr 2016

**General Comments**

This paper presents the results of a super-ensemble particle tracking simulation of the debris originating from the probable crash site of the MH370. The paper is clear and easy to read. It is acceptable after a minor revision. I suggest enriching it as follows:

1) In the abstract, you should add a better statement of the scientific problem you are addressing and why your approach is general (or can be generalized) and applicable to any area and to different cases/objects at sea.

2) In the Introduction, it is missing a paragraph to situate this work in the context of (i) trajectory modelling studies, with particular attention to previous works about tracking objects/debris at sea, (ii) ensemble and super-ensemble techniques (both a general overview and a specific focus to trajectory modelling applications). As you may find

in the state-of-the-art literature, the ocean model horizontal resolution, the temporal frequency of currents and the inclusion of the wave effect in the transport of objects at sea might be important. I suggest adding some comments on these issues in the methodology and results sections and comparing your approach with previous studies.

3) A paragraph about the ocean general circulation in the area and how it is connected with the probability density patterns is missing. I suggest adding this in the results section to enrich your findings.

4) Please consider including in your revised version the specific comments listed below.

**Specific comments**

Page 1 Line 7: "to improve" might not be the proper verb to use; you cannot run the super-ensemble without at least one observation, right? So, you are not improving your simulation, it is absolutely necessary to have at least the position of the discovery of the debris on Réunion.

Page 1 Line 9: "initial probability density": I would add of "marine debris distribution". Please specify "initial" at which date/time corresponds.

Page 1 Line 9: "current" are the underwater research still on-going?

Page 1 Line 9-10: "later times" add specific time frame (date).

Page 2 Eq (1): Have you used any existing trajectory model or did you build your own?

Page 2 Line 20: I suggest adding more on "the wind drag coefficient": 1) what are the physical processes this term is accounting for (you should cite here the literature on leeway factors)? 2) please explain here or at page 2 Line 27 why these coefficient are not accurately known.

Page 2 Eq (2): Is $\Delta t$ in Eq (2) different from dt in Eq (1)?

Page 2 Line 24: Could you add more about the choice of the diffusivity? It may be

depending on the ocean model resolution you are using (reference to De Dominicis et al. 2012 for the original work about diffusivity).

Page 3 Eq (4). Does i go from 1 to Nk?

Page 3 Line 6: "certain distance" is vague; please specify values and/or reasons to choose a certain value.

Page 3 Line 12: Could you explain the choice of 100 km and 3 months?

Page 3 Eq. (6): Is this a standard choice?

Page 3 Line 15: Is t=90 days somehow related with the 3 months of Line 12? What happens to your model results if you choose different values?

Page 3 Line 19-21: Wind drag coefficients are not initial conditions. I suggest enlarging and moving this sentence to Section 2.1. Does the wind drag depend only on the size of the object or also on the in/out water parts of the objects? Is the limit of 2.5% adequate? Please add more on the uncertainty of the size and properties of the objects in this specific case.

Page 3 Line 21: Were the 5000 particles released in a single point or on an area?

Page 3 Line 25: Are you using forecast or analysis? Please specify. Have the surface currents you are using been validated (in other studies, please cite)?

Page 4 Line 2: Why were you using daily currents at the very beginning of the simulation?

Page 4 Line 10: "drifters" were not mentioned before, better to change to particles.

Page 4 Line 12-13: Is there a way to calculate the super-ensemble without the weights? Otherwise it's not clear the sense of this sentence.

Page 4 Line 17: In the figures you are not showing September 2014 but August 2014, please cite your figures and make consistent the text with the figures.
[Figure]

Page 4 Line 18: In the figures you are not showing October 2014 but November 2014, please cite your figures and make consistent the text with the figures. In general, I like the figures, but I do not understand the criteria to show one month instead of another, they are not at regular intervals, nor linked with the comments in the text.

Page 4 Line 19: Add the reference to Figure 3. Can you comment on the probability patterns that are shown in Figure 3? Top left panel shows very low probability around Réunion Island and the probability density has a particular shape, narrowing around the Réunion Island. Could you comment on this? Bottom panels show hot spots of higher probability around the Réunion Island. I guess this is because the piece of the wing was found in August 2015 (why don't you show the probability density of August 2015?). Do these hot spots really identify higher probability density areas or is it just an artefact due to the fact that you have just one observation? Some comments to link these patterns with the methodology you used would strengthen your manuscript.

Page 5 Line 4-5: "satellite data analysis" not clear. Did you mean the final satellite communication? If so, this information was used to "determine the underwater search area" and it is not independent from it (so you can say that is in agreement with just on of those information).

Page 5 Line 5-6: Does "search area" refer to the wider or the underwater search area?

Page 5 Line 13: Why in a conventional ensemble adding additional information would be very difficult? This is one of the strength of your paper, please add more on this (I would suggest to add this also in the abstract).

Page 5 Line 14-15: This statement is another strength of your paper. I suggest commenting more on this. In my opinion, you could insert the appendix as another section of the paper and I propose changing the title in order to highlight that the main strength of that section is to show the capability of easily including more information, rather than "Mozambique debris". In general, you should focus on the scientific value of the methodology that you are presenting rather than on the "chronicle" of the accident.

[Figure]

Page 6 Line 25: Why in this case did you choose 50 km? Was 100 km in the previous case? The choice of this parameter should not be arbitrary and should be clarified.

**Technical corrections**

Title: "supersensemble" to "superensemble"

page 2 line 8: "presents" to "present

---

## Author Comment (AC1) · 1 Jul 2016

**Dear reviewer,**

**Thank you for reviewing our paper and for your kind words. Please find our response to your questions and comments inline below.**

*This manuscript is a very important contribution to Natural-Hazards-And-Earth-System-Sciences and is entirely suitable for publication. The paper adds new knowledge to the overall body of scientific understanding regarding particle tracking in the environment. The authors, especially the second and third ones, are very well recognized and highly regarded researchers in the subject area of this manuscript. The authors build on their previous work with lagrangian modeling and employ well established meteorological (ecmwf) and oceanographic (European Copernicus Marine Envi-*

*ronment Monitoring Service) data sets in their analysis. The use of drift simulations will take on especial importance as air and sea travel continues to rise and world events become more dangerous putting more of our population at risk. The research results presented are highly original. The approach taken in the conduct of this research will set a standard for future drift modeling.*

*I believe the paper is free of errors in logic; their case is very well presented and made.*

*Some comments: 1. I think that the appendix should be moved into the main body of the manuscript. It is possible that the Mozambique debris may not be related to MH370. But the method of including it in the analysis is important for the readers. In fact there are other announcements of possible debris. It would be so good to include them all.*

*http://www.ibtimes.com/flight-mh370-update-chinese-vessel-resume-search-recover-lost-towfish-amid-suspected-2350977   http://www.livescience.com/54158-debris-found-from-mh370-malaysia-plane.html   http://www.inquisitr.com/2941135/malaysia-airlines-flight-mh370-new-debris/  http://www.inquisitr.com/2968872/malaysia-airlines-flight-mh370-inside-plane-debris/  http://www.inquisitr.com/2957754/malaysia-airlines-flight-mh370-planes-interior-debris/*

**We agree that the appendix should be included in the main body of the paper. Especially since the debris found in Mozambique has now been confirmed to be part of the horizontal stabiliser of the missing aircraft.**

**Furthermore, if the editor agrees, we would like to extend the simulation until May 2016 and include an additional 3 debris discoveries that have been confirmed in the meantime:**

**- Flap fairing, Guinjata Bay, Mozambique (December 2015)**
**- Piece of engine cowling, Mossel Bay, South Africa (December 2015)**
**- Interior panel, Rodrigues Island, Mauritius (March 2016)**

This would include all of the currently confirmed discoveries and we believe it adds to the impact of this manuscript at the time of publication.

*2. It would be interesting if the authors could discuss a backtracking approach. Release particles from Reunion Island and see where they came from. Play the currents back in time.*

**We have examined the possibility of using a backtracking approach, but we found that the results are not accurate enough to provide meaningful information. The reason is that in a forward simulation a particle moves from x0 to x1 using the forcing fields at x0, while in a backward simulation the same particle returns from x1 to x0 using the opposite of the fields at x1. The assumption that x0 and x1 are close enough such that the field values are the same is only approximately true, introducing a small error in the backtracking step. As the output of one integration step is the input for the next, these errors accumulate and become noticeable for simulations longer than a couple of weeks.**

*3. Why is the approach called a "superensemble" as opposed to an ensemble? What makes is super?*

**Superensemble refers to the fact that the superensemble is an ensemble of ensembles.**

*4. It seems to me that there has been much published about the debris field. Has all the literature been reviewed for this manuscript?*

**Various groups have performed simulations of the debris field, but unfortunately none of these studies have been published. Without a full description of the method and data that were used it is difficult to make an in-depth comparison. Based on the available information the forward simulations agree with our current result. The application of the superensemble technique has not been done in any of the unpublished studies.**

**http://www.marine.csiro.au/ griffin/MH370/**
**http://iprc.soest.hawaii.edu/news/MH370_debris/IPRC_MH370_News.php**
**http://www.geomar.de/n3972-e**
**http://www.geomar.de/n4432-e**

*5. I like the figures. I think Figure 1 should contain labels for Mozambique and Madagascar. Others?*

**Thank you. Figure 1 will be updated to include all the locations where debris has been discovered.**

---

## Author Comment (AC2) · 1 Jul 2016

**Dear reviewer,**

**Thank you for reviewing our paper and for your detailed comments. Please find our response below.**

*General Comments This paper presents the results of a super-ensemble particle tracking simulation of the debris originating from the probable crash site of the MH370. The paper is clear and easy to read. It is acceptable after a minor revision. I suggest enriching it as follows:*

*1) In the abstract, you should add a better statement of the scientific problem you are addressing and why your approach is general (or can be generalized) and applicable*

[Figure]

*to any area and to different cases/objects at sea.*

**We propose to add to the abstract: Multiple model realisations are used with different starting locations and wind-drag parameters. The model realisations are combined into a superensemble, adjusting the model weights to best represent the discovered debris. The chosen approach can be easily generalised to other drift simulations where observations are available to constrain unknown input parameters.**

*2) In the Introduction, it is missing a paragraph to situate this work in the context of (i) trajectory modelling studies, with particular attention to previous works about tracking objects/debris at sea, (ii) ensemble and super-ensemble techniques (both a general overview and a specific focus to trajectory modelling applications). As you may find in the state-of-the-art literature, the ocean model horizontal resolution, the temporal frequency of currents and the inclusion of the wave effect in the transport of objects at sea might be important. I suggest adding some comments on these issues in the methodology and results sections and comparing your approach with previous studies.*

**We propose to extend the introduction as follows: The use of numerical simulations for search-and-rescue modelling dates back to the early 1970's, when the U.S. Coast Guard introduced its Computer Assisted Search Planning system [Frost and Stone, 2001]. Developments in meteorology, oceanography and computer technology have since led to large improvements in the performance and accuracy of the numerical models. However, the basic methodology of drift modelling has remained the same [Hackett et al., 2006, Breivik et al., 2012], using an ensemble of particles that respond linearly to the wind speed. The relationship between drift and wind speed has been the subject of several decades worth of field experiments, ultimately leading to a system of standardised coefficients or leeway factors [Allen and Plourde, 1999].**

**The use of superensembles, a weighted combination of multiple models, was**

first introduced in meteorology by Krishnamurti et al. (1999) as a method to create multimodel forecasts. Applications in oceanograpy and more specifically in surface drift modelling [Rixen et al., 2007, 2008] have so far focused on improving the forcing fields used as input to the drift model. This paper presents a slightly different approach, applying the superensemble technique to multiple realisations of the drift model itself. Using long term global ocean and weather reconstructions, the superensemble drift model is used to produce probability distributions for the surface drift of aircraft debris originating from an accident in the southern Indian Ocean.

With additional references:

Frost, J. R. and Stone, L. D.: Review of Search Theory: Advances and Applications to Search and Rescue Decision Support, Tech. Rep. CG-D-15-01, U.S. Coast Guard Research and Development Center, 2001

Hackett, B., Breivik, Ø., and Wettre, C.: Forecasting the drift of objects and substances in the ocean, Ocean Weather Forecasting, pp. 507-523, 2006

Breivik, Ø., Allen, A. A., Maisondieu, C., and Olagnon, M.: Advances in Search and Rescue at Sea, Ocean Dyn, 63(1), pp. 83-88, 2012

Rixen, M. and Ferreira-Coelho, E.: Operational surface drift prediction using linear and non-linear hyper-ensemble statistics on atmospheric and ocean models, Journal of Marine Systems, 65(1-4), pp. 105 - 121, 2007

Rixen, M., Ferreira-Coelho, E., and Signell, R.: Surface drift prediction in the Adriatic Sea using hyper-ensemble statistics on atmospheric, ocean and wave models: Uncertainties and probability distribution areas, Journal of Marine Systems, 69(1-2), pp. 86-98, 2008

*3) A paragraph about the ocean general circulation in the area and how it is connected with the probability density patterns is missing. I suggest adding this in the results*

*section to enrich your findings.*

**We propose to add to the results section: The results show the counter-clockwise pattern of the Indian Ocean Gyre, with the West Australian Current moving north along the Australian coast and the westward South Equatorial Current around 20°S.**

*4) Please consider including in your revised version the specific comments listed below.*

**Thank you for your detailed comments. The textual suggestions will be taken into account. Please find our answers to specific questions below.**

*Specific comments Page 1 Line 7: "to improve" might not be the proper verb to use; you cannot run the super-ensemble without at least one observation, right? So, you are not improving your simulation, it is absolutely necessary to have at least the position of the discovery of the debris on Réunion. Page 1 Line 9: "initial probability density": I would add of "marine debris distribution". Please specify "initial" at which date/time corresponds.*

*Page 1 Line 9: "current" are the underwater research still on-going?*

**The underwater search is still ongoing. An area of 120,000 square kilometres is planned to be searched of which about 105,000 has been completed. See the operational updates of the ATSB in Australia: https://www.atsb.gov.au/mh370-pages/updates/operational-update/**

*Page 1 Line 9-10: "later times" add specific time frame (date).*

*Page 2 Eq (1): Have you used any existing trajectory model or did you build your own?*

**The trajectory model used for this study is a new model that has been developed at CMCC by the authors.**

*Page 2 Line 20: I suggest adding more on "the wind drag coefficient": 1) what are the physical processes this term is accounting for (you should cite here the literature on*

*leeway factors)? 2) please explain here or at page 2 Line 27 why these coefficient are not accurately known.*

**We propose to add: The constant L is the wind drag coefficient or downwind leeway [Allen and Plourde, 1999] of the object. It accounts for both direct and indirect (e.g. Stokes drift) wind-induced motion of a particle.**

*Page 2 Eq(2): Is $\Delta t$ in Eq(2) different from dt in Eq(1)?*

**The RK4 method involves several increments of dt to calculate the displacement in one model integration timestep $\Delta t$. The noise term is added after each model integration step. (As the noise terms are uncorrelated normally distributed variables with $\sigma \propto \sqrt{\Delta t}$, the total displacement due to turbulent diffusion in a given time interval is independent of the choice for $\Delta t$.)**

*Page 2 Line 24: Could you add more about the choice of the diffusivity? It may be depending on the ocean model resolution you are using (reference to De Dominicis et al. 2012 for the original work about diffusivity).*

**The choice of $K_h$ has only minimal impact on the results of the simulation, therefore we chose to adopt the value from a well-established operational model. We believe that an in-depth discussion of diffusivity is beyond the scope of this paper.**

*Page 3 Eq(4). Does i go from 1 to Nk?*

**Correct, this will be added to Eq. (4)**

*Page 3 Line 6: "certain distance" is vague; please specify values and/or reasons to choose a certain value.*

**This distance is not defined at this point as it is a choice that depends on the circumstances.**

*Page 3 Line 12: Could you explain the choice of 100 km and 3 months?*

The timespan of 3 months represents the fact that the aircraft part may have washed up some time before it was discovered. The distance of 100km is half of the distance to the neighbouring island of Mauritius. The results do not show a strong dependence on this choice. However, if the distance is chosen too small many of the model realisations may be completely excluded from the combination (receive a weight of 0).

*Page 3 Eq. (6): Is this a standard choice?*

The ensemble of one model realisation can be considered as a single value (the ensemble mean) and an uncertainty/error (the ensemble spread). The prior likelihood of Eq. (6) is equivalent to using the error weighted average of the different model realisations for the superensemble.

*Page 3 Line 15: Is t=90 days somehow related with the 3 months of Line 12? What happens to your model results if you choose different values?*

The time of 90 days is chosen to make sure that the particles of each model realisation have had enough time to spread out, but the particles have not yet started to accumulate on land. This gives an optimal measure of the variance of the different realisations. The results of the simulation show little to no dependence on this choice as long as the above considerations are taken into account. There is no relation to the 3 months used for the Réunion discovery.

*Page 3 Line 19-21: Wind drag coefficients are not initial conditions. I suggest enlarging and moving this sentence to Section 2.1. Does the wind drag depend only on the size of the object or also on the in/out water parts of the objects? Is the limit of 2.5% adequate? Please add more on the uncertainty of the size and properties of the objects in this specific case.*

This section describes how the superensemble is constructed, as such it is better to describe the location and wind drag coefficients together. The wind drag

**depends on the shape and in/out of water parts of the object. The aircraft parts capable of floating for long periods of time are generally hollow panels that float horizontally on the surface. We believe that an upper limit of 2.5% is reasonable, as aircraft debris with large out of water parts would eventually break up into smaller pieces.**

*Page 3 Line 21: Were the 5000 particles released in a single point or on an area?*

**Particles were released following a Gaussian distribution around the arc of the final communication. The standard deviation of this Gaussian is set to 50km, in order to populate the entire wide search area.**

*Page 3 Line 25: Are you using forecast or analysis? Please specify. Have the surface currents you are using been validated (in other studies, please cite)?*

**The name of the dataset refers to the fact that it contains both analysis of historical data and short term forecasts in the same format. The historical data used in this paper correspond to analysis only. We will remove "and forecast" from the text to avoid confusion.**

*Page 4 Line 2: Why were you using daily currents at the very beginning of the simulation?*

**Data with two-hourly temporal resolution were not available for the first days of the simulation.**

*Page 4 Line 10: "drifters" were not mentioned before, better to change to particles.*

*Page 4 Line 12-13: Is there a way to calculate the super-ensemble without the weights? Otherwise it's not clear the sense of this sentence.*

**The superensemble could be used without weights or by giving equal weight to each model realisation. In that case it is essentially a conventional ensemble with particle parameters sampled from all possible values. This is what is generally**

**used in for example search-and-rescue models where no additional information is available.**

*Page 4 Line 17: In the figures you are not showing September 2014 but August 2014, please cite your figures and make consistent the text with the figures. Page 4 Line 18: In the figures you are not showing October 2014 but November 2014, please cite your figures and make consistent the text with the figures. In general, I like the figures, but I do not understand the criteria to show one month instead of another, they are not at regular intervals, nor linked with the comments in the text.*

**Figures will be updated to be consistent with the times mentioned in the text.**

*Page 4 Line 19: Add the reference to Figure 3. Can you comment on the probability patterns that are shown in Figure 3? Top left panel shows very low probability around Réunion Island and the probability density has a particular shape, narrowing around the Réunion Island. Could you comment on this? Bottom panels show hot spots of higher probability around the Réunion Island. I guess this is because the piece of the wing was found in August 2015 (why don't you show the probability density of August 2015?). Do these hot spots really identify higher probability density areas or is it just an artefact due to the fact that you have just one observation? Some comments to link these patterns with the methodology you used would strengthen your manuscript.*

**It is of course true that weighting the model realisations based on the discovery of debris on Réunion increases the probability density in this location. However, it is important to realise that the superensemble consists of 30 plausible simulations and the weighting only modifies the linear combination of simulations that is chosen.**

**Most particles eventually enter into the eastward current around 20°S, but due to the different starting locations they do so at different times. This causes the density patterns that you see.**

*Page 5 Line 4-5: "satellite data analysis" not clear. Did you mean the final satellite communication? If so, this information was used to "determine the underwater search area" and it is not independent from it (so you can say that is in agreement with just on of those information).*

**Satellite data analysis refers to the work done by Ashton et al. 2015. In addition to determining the arc of the final satellite communication, the analysis also calculated a most probable flight path using the assumption that it was flying with constant speed and direction. The underwater search area is based on this path. While the wide search area is used as input to the simulation, the underwater search area is not. The agreement in this case refers to the agreement of the simulation with the underwater search area.**

*Page 5 Line 5-6: Does "search area" refer to the wider or the underwater search area?*

**It refers to the wide search area, this will be clarified.**

*Page 5 Line 13: Why in a conventional ensemble adding additional information would be very difficult? This is one of the strength of your paper, please add more on this (I would suggest to add this also in the abstract).*

*Page 5 Line 14-15: This statement is another strength of your paper. I suggest commenting more on this. In my opinion, you could insert the appendix as another section of the paper and I propose changing the title in order to highlight that the main strength of that section is to show the capability of easily including more information, rather than "Mozambique debris". In general, you should focus on the scientific value of the methodology that you are presenting rather than on the "chronicle" of the accident.*

**The appendix was added in the final stage of preparing the paper and at the time there were many doubts whether the debris found in Mozambique was related to flight MH370. It will be added into the main body of the final revised paper.**

*Page 6 Line 25: Why in this case did you choose 50 km? Was 100 km in the previous*

[Figure]

*case? The choice of this parameter should not be arbitrary and should be clarified.*

**This was in fact a typo and in both cases 100km is used.**

*Technical corrections*

*Title: "supersensemble" to "superensemble" page 2 line 8: "presents" to "present*

**The title is a rather embarrassing place to make a typo, thank you for catching this.**